# Climate change alters the future of natural floristic regions of deep evolutionary origins

Samuel Minev-Benzecry ● & Barnabas H. Daru ● ✉

Biogeographic regions reflect the organization of biotas over long evolutionary timescales but face alterations from recent anthropogenic climate change. Here, we model species distributions for 189,269 vascular plant species of the world under present and future climates and use this data to generate biogeographic regions based on phylogenetic dissimilarity. Our analysis reveals declines in phylogenetic beta diversity for years 2040 to 2100, leading to a future homogenization of biogeographic regions. While some biogeographic boundaries will persist, climate change will alter boundaries separating biogeographic realms. Such boundary alterations will be determined by altitude variation, heterogeneity of temperature seasonality, and past climate velocity. Our findings suggest that human activities may now surpass the geological forces that shaped floristic regions over millions of years, calling for the mitigation of climate impacts to meet international biodiversity targets.

Biogeographic regionalization is the grouping of regions based on their shared elements such as taxonomic composition[1,2] or vegetation features[3]. This classification forms the basis of many biogeographic, conservation, and macroecological investigations[4,5]. Incorporating phylogenetic information into regionalization analysis, i.e., phylogenetic regionalization, can reveal new insights into the ecological processes that structure biodiversity over geologic time scales, such as vicariance, dispersal, speciation, extinction, and niche conservatism[6–8]. Within a biogeographic region, change in species composition (β-diversity) is expected to be relatively similar because of shared diversification and colonization history[6,9] but separated from neighboring regions by biogeographical boundaries[10,11]. The boundaries are shallow if they separate species assemblages with limited dissimilarity (floristic regions henceforth) and deep if they separate highly dissimilar species assemblages (realms henceforth)[12]. Concomitantly, the pronounced impact of human-induced global change on biodiversity[13–15] poses a threat to biogeographic regions which have their origins in deep evolutionary time[7,16]. The effects may be severe when they affect primary producers like vascular plants, because the extirpation or change in plant species assemblages can simplify and disrupt ecosystem functioning[17,18].

Studies investigating changes in biogeographic regions have focused on the role of invasive species introductions and local extirpations of animal taxa and have revealed that alien species alter biogeographic regions[19–21]. By contrast, a recent study[22] suggested that extinctions exert a greater homogenizing effect on plant biogeographic regions than introductions. However, this conclusion was drawn from the compilation of regional checklists and floras aggregated to artificial and coarse administrative units such as countries or provinces[22]. Although anthropogenic climate change has been identified as one of the key drivers of biodiversity change[23,24], no studies have investigated how future climate change would impact plant biogeographic regions. Climate change can influence plant species distribution by exceeding critical thermal limits, or indirectly by determining the availability of key resources such as water and nutrients[25]. As a result of such anthropogenic filtering, species may undergo range shifts in various directions in response to changes in environmental conditions to maintain equilibrium with suitable living conditions[26,27]. These shifts can result in local extirpations or introductions in previously unoccupied areas, leading to changes in biotic composition (β-diversity) across biogeographic regions, with negative consequences for ecosystem services such as primary production[28,29].

We hypothesize that if species' climatically suitable habitats contract under worsening climate conditions, β-diversity will likely increase, leading to the compositional differentiation of biogeographic regions[30] (Fig. 1). Conversely, in areas where species ranges expand to colonize new climatically suitable areas, β-diversity will decrease, and the composition of species assemblages will experience homogenization across geographic space[6]. Climate change can also

Department of Biology, Stanford Universitys, Stanford, CA, USA. ✉ e-mail: bdaru@stanford.edu

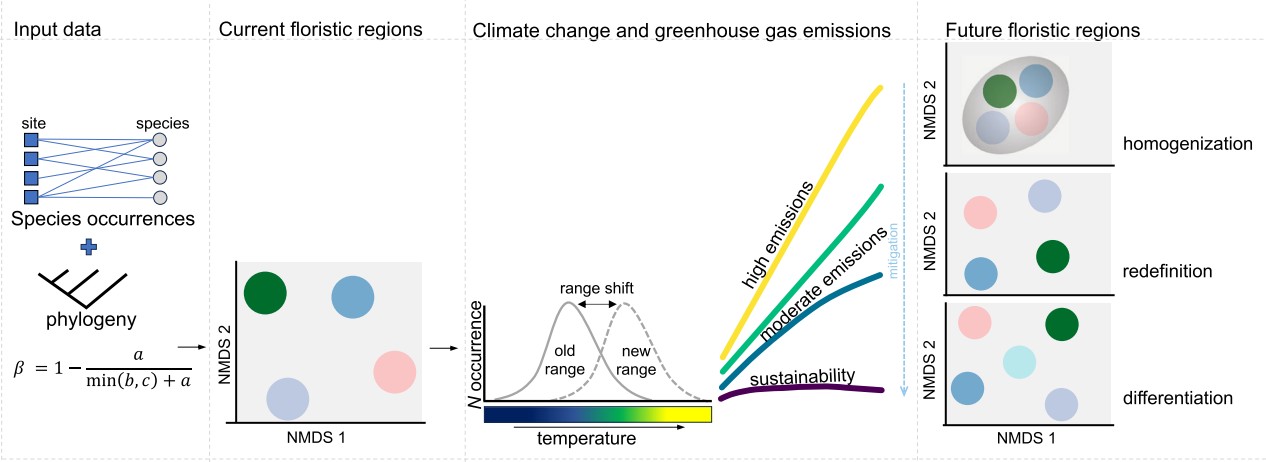

**Fig. 1 | Hypotheses for how climate-induced biodiversity change can alter the future of floristic regions.** Clusters of present floristic regions in non-metric multidimensional scaling (NMDS) space defined by clustering species occurrences with phylogenetic information and applying pairwise Simpson's β-diversity, $\beta_{sim}$, where $a$ is the number of shared species between two grid cells, and $b$ and $c$ are the numbers of species unique to each grid cell. As climate changes due to human impacts, species may undergo range shifts in response to environmental factors such as temperature to maintain equilibrium with suitable living conditions. Consequently, and depending on the climate scenarios over the 21st century, we predict that climate change can homogenize, redefine or differentiate the future of floristic regions.

redefine biogeographic boundaries by rearranging species distributions without changes in their number[20]. Altogether, we predict that climate change can homogenize, differentiate, and redefine biogeographic regions resulting in different biogeographic regions than we see today (Fig. 1).

Here, we assess how anthropogenic climate change could homogenize, differentiate, and redefine natural biogeographic regions of vascular plants. We compare biogeographic regions delineated with species distributions in the present with those delineated based on modeled distributions under future climate scenarios. Specifically, we used species distribution models to analyze 454 million occurrence records for vascular plants from herbaria and observations resulting in individual species-level native range maps for 189,269 species under present and future climatic projections throughout the twenty-first century. The selection of the 189,269 species reflects those with successfully modeled distributions that are consistent across different time horizons and climate scenarios. Future projections are based on the Model for Interdisciplinary Research on Climate v.6 (MIROC6)[31] and four Shared Socioeconomic Pathways (SSP 126, 245, 370 and 585). These pathways represent varying levels of climate mitigation, ranging from best-case (SSP126) to moderate (SSP245 and SSP370) and high emissions (SSP585) scenarios[32,33]. Specifically, we modeled current plant distributions as a function of current environmental variables and used this model to predict future plant distributions at new values of climate under different future scenarios for T1: 2021-2040, T2: 2041-2060, T3: 2061-2080, and T4: 2081-2100. For each time horizon, we combined the modeled distributions with a species-level phylogeny of plants and used pairwise phylogenetic β-diversity to generate biogeographic regions and address the following questions: (1) How might climate change homogenize, differentiate, or redefine plant biogeographic regions? (2) How would climate change alter plant biogeographic boundaries? and (3) What are the determinants of plant biogeographic boundaries in the age of human impact? We uncover patterns of both differentiation and homogenization within existing biogeographic regions that can lead to persistence of some biogeographic boundaries, while climate will alter deeper biogeographic boundaries that separate historically distinctive plant assemblages. This suggests that human activities may now be a dominant force structuring plant biogeographic regions, potentially overriding the influence of evolutionary history.

## Results and discussion

### Climate change and the alteration of biogeographical patterns

We used niche-based distribution models to delineate floristic regions for plant species across four future time periods (T1–T4) using MIROC6 (a climate model for representing various processes of the Earth's climate system)[31] and four Shared Socioeconomic Pathways (SSPs). Using the minimum number of clusters explaining 85% (corresponding to floristic realms) and 90% (regions) of Simpson's phylogenetic dissimilarity based on the unweighted pair group method with arithmetic mean (Supplementary Fig. 1) as in previous studies[4,7–9,12], we identified 19 distinct clusters of 100 km × 100 km grid cells which we define as floristic regions nested within 10 highly distinctive clusters (floristic realms) in the present-day (Fig. 2). The number of clusters for the present-day floristic regions were consistent with future floristic regions for some climate scenarios and different for others, ranging from 21 regions in timeframe T1 (2021–2040) to 25 by 2100 (Supplementary Fig. 2). We used these cutoffs to map plant biogeographic regions for future distributions (Fig. 2). Our delineated plant biogeographic regions in the present-day show moderate spatial correlation with future biogeographic regions, and slightly lower on average compared to the correlation among future biogeographic regions for most time periods (Supplementary Fig. 3). Future projections indicate the Circumboreal, Afrotropics, Saharo-Arabian, Malesian, and Indian-Indochinese regions are similar to present-day floristic regions (Fig. 2 and Supplementary Fig. 4) and consistent with established floristic regions[7,8]. Differences in our projected floristic regions include the merging of Amazonian and Brazilian floristic regions into a single unit in future time periods (T3 and T4), along with the disappearance of Californian floristic region during the same period (Supplementary Fig. 4). These findings are projected to remain consistent across all SSP climate scenarios (Supplementary Fig. 5). These findings suggest an initial differentiation and redefinition of biogeographic regions in the mid-century, followed by coalescence of regions towards the end of the century, supporting our hypothesis that climate change can both differentiate and redefine some plant biogeographic regions.

To assess changes in the composition of biogeographic regions, we measured shifts in phylogenetic beta diversity ("phylobeta diversity")—the standard metric commonly used to delineate biogeographic regions[4,12]—across future projections in grid cells relative to present-day floristic regions. We found that future phylobeta diversity is

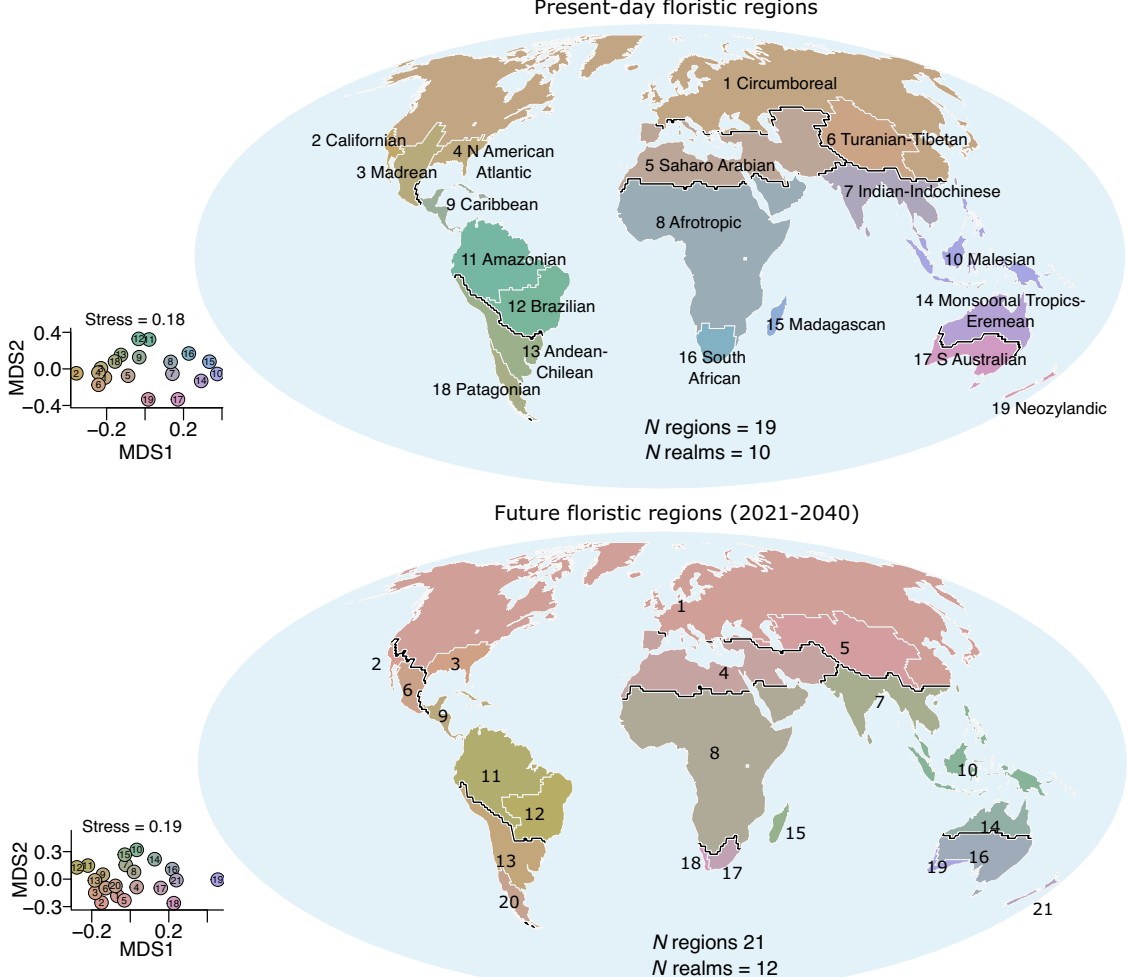

**Fig. 2 | Changes in vascular plant biogeographic regions under current and future climate scenarios in geographic space and in non-metric multidimensional scaling (NMDS) ordination space.** Top, global floristic regions in the present-day delineated by clustering modeled range maps for 189,269 vascular plant species with phylogenetic information and applying pairwise Simpson's β-diversity between 100 km × 100 km grid cells. Bottom, future floristic regions. Future species distributions were predicted by first modeling current plant species distributions as a function of current environmental variables and then using this model to predict future plant distributions at new values of climate under different future scenarios and then using that to generate floristic regions for T1: 2021-2040 based on the mean of pairwise distance matrices of phylogenetic β- diversity for the four shared socioeconomic pathways (SSP126, SSP245, SSP370, and SSP585). See supplementary information for other time periods T2: 2041-2060, T3: 2061-2080, and T4: 2081-2100. The colors in the map and NMDS plots are identical and indicate levels of differentiation of the flora in different floristic regions such that floristic regions with similar colors have similar clades and those with different colors differ in the plant clades they enclose. Black lines separate floristic realms, while white lines separate floristic regions. The numbers in the map and NMDS plots are arbitrary and meant for visual reference to identify clusters for each time period and do not represent a one-to-one match across time periods. The maps are in the equal-area World Mollweide projection. Source data are provided as a Source Data file.

projected to be relatively lower when considering phylobeta diversity within present-day biogeographic regions, with a global mean reduction in Cohen's $d$ effect size of -0.0058 in T1 to -0.06 in T4 (2081-2100) (Fig. 3, Supplementary Table 1). Phylobeta diversity is predicted to decrease the most in the Circumboreal, North American Atlantic, Madagascan, Monsoonal Tropics-Eremean, and Neozylandic floristic regions ($P < 0.05$, Cohen's d effect size; Fig. 3), and this will intensify toward the end of the century in T4 (2081-2100). The projected decline of phylobeta diversity within floristic regions supports our hypothesis that climate change can homogenize plant biogeographic regions.

Our findings illustrating similarities and differences among present-day and future regionalizations are consistent at the floristic realm level and across different dissimilarity metrics of phylogenetic beta diversity (Simpson's and Sorensen indices; Supplementary Fig. 6–8), highlighting the robustness of our findings. While previous studies have demonstrated the influence of non-native introductions and species extirpations in altering biogeographical patterns for terrestrial vertebrates[20], snails[19], and fish[21], our study is the first global assessment of how climate change alone alters biogeographical patterns, reinforcing recent findings that suggest a stronger role of climate change compared to land use change in impacting biodiversity[34].

## Shifts in biogeographical boundaries under climate change

We evaluated shifts in region and realm boundaries as the boundaries between biogeographic regions, at a grain resolution of 100 km. We found that all boundaries are distributed across all continents (Fig. 4) consistent with the boundaries of previous global regionalizations of plants[7,8] and tetrapods[12,35–37]. Some present-day floristic boundaries overlap with projected future floristic boundaries, particularly in the Andean (T1-T4), Tibetan-Hengduan-Himalaya (T1-T4), and Rocky Mountains (T1, T2, and T4) regions (Fig. 4), suggesting that the factors currently separating distinct plant communities will likely continue to act as barriers as the climate changes. However, in the Brazilian-Amazonian, South African, Afrotropics (Horn of Africa) and Australasian

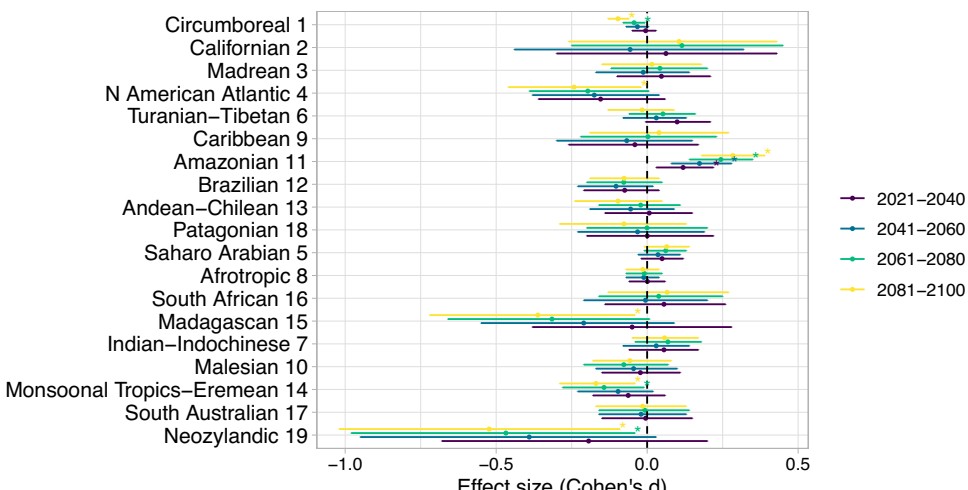

**Fig. 3 | Changes in phylogenetic beta diversity under climate change relative to present-day floristic regions.** The magnitude of change in Simpson's β-diversity across spatial and temporal scales was assessed by comparing the grid-cell compositional dissimilarity for delineating present vs future floristic regions when considering Simpson's β-diversity within present-day floristic regions. A two-sided $t$-test was used to assess differences, followed by Cohen's $d$ with 1000 bootstrap replicates to estimate effect size. Data are presented as Cohen's $d$ ranging from 0 (no effect) to +1 or −1 (large effect), with positive values indicating differentiation, whereas negative values indicate homogenization. The error bars indicate 95% confidence intervals, and the statistical significance of the $t$-test are indicated with asterisks ($P < 0.01$). Source data are provided as a Source Data file.

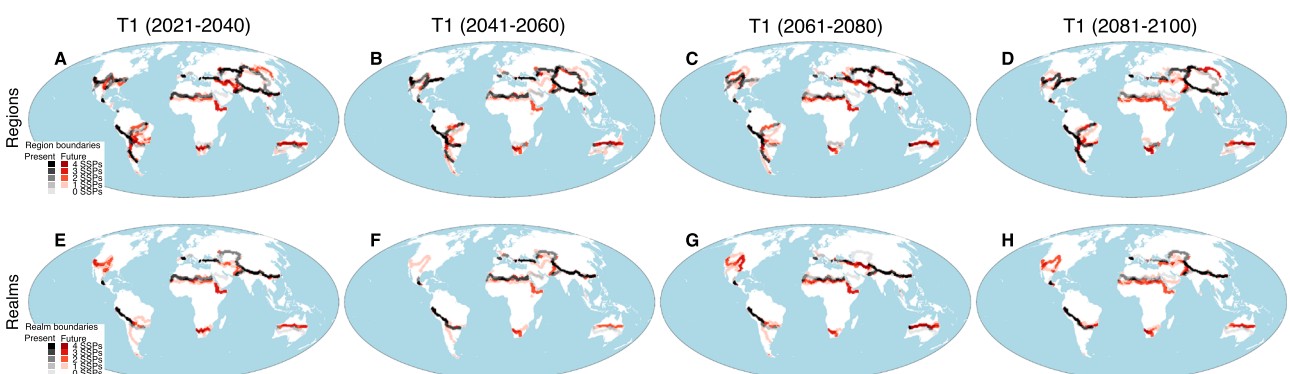

**Fig. 4 | Changes in the boundaries of present-day floristic regions and realms in comparison with future boundaries.** Top row: all region boundaries for present-day and future floristic regions indicating areas of congruence and incongruence of boundaries across climate scenarios for (**A**) T1: 2021-2040. **B** T2: 2041-2060. **C** T3: 2061-2080. **D** T4: 2081-2100. Bottom row: realm boundaries for present-day and future floristic realms indicating areas of congruence and incongruence of boundaries across climate scenarios for (**E**) T1: 2021-2040. **F** T2: 2041-2060. **G** T3: 2061-2080. **H** T4: 2081-2100. The maps are in the equal-area World Mollweide projection. Source data are provided as a Source Data file.

floristic regions, future boundaries will increasingly diverge from present boundaries by the end of the century, representing mismatches between present and future boundaries (Fig. 4A–D).

These differences are further reflected in the spatial correlation of present-day and future boundaries (Supplementary Fig. 9) where present-day region boundaries show a weaker correlation with projected future region boundaries combined across climate scenarios (Pearson's $r = 0.79$ to 0.80, $P \ll 10^{-10}$) compared to the correlation among individual scenarios of future region boundaries for periods T2-T4 ($r = 0.86$ to 0.89, $P \ll 10^{-10}$, Supplementary Fig. 9, Supplementary Table 2). For floristic realms, our projected realm boundaries showed pronounced incongruence to boundaries of present-day floristic realms compared to those observed for all region boundaries. Present-day realm boundaries are weakly correlated with future realm boundaries (Pearson's $r = 0.54$ to 0.63, $P \ll 10^{-10}$) compared to the correlation among future realm boundaries (Pearson's $r = 0.66–0.82$, Supplementary Fig. 9 and Supplementary Table 3). Several of these projected realm boundaries that do not overlap present-day

boundaries include Southern Africa, Australasia, and Central United States (Fig. 4). These findings suggest that climate change is likely to alter deep floristic boundaries as opposed to shallow boundaries, although in our context, deep boundaries correspond to the separation of highly dissimilar species assemblages, and not necessarily deep evolutionary times. The tendency for the alteration of deeper floristic boundaries under climate change might reflect the pattern of historical extinctions in these realms, with surviving species representing remnants of once much more diverse clades[38]. It could also mean that floristic realms may be getting less suited to rapid climate shifts in the age of human impact, heightening the risk of boundary alterations.

### Determinants of biogeographical boundaries

We tested whether the positions of present and future plant biogeographic boundaries are driven by areas that have experienced past climate change, tectonic movements, temperature seasonality, precipitation seasonality, or variations in orographic barriers as these metrics are hypothesized to influence biogeographic boundaries in

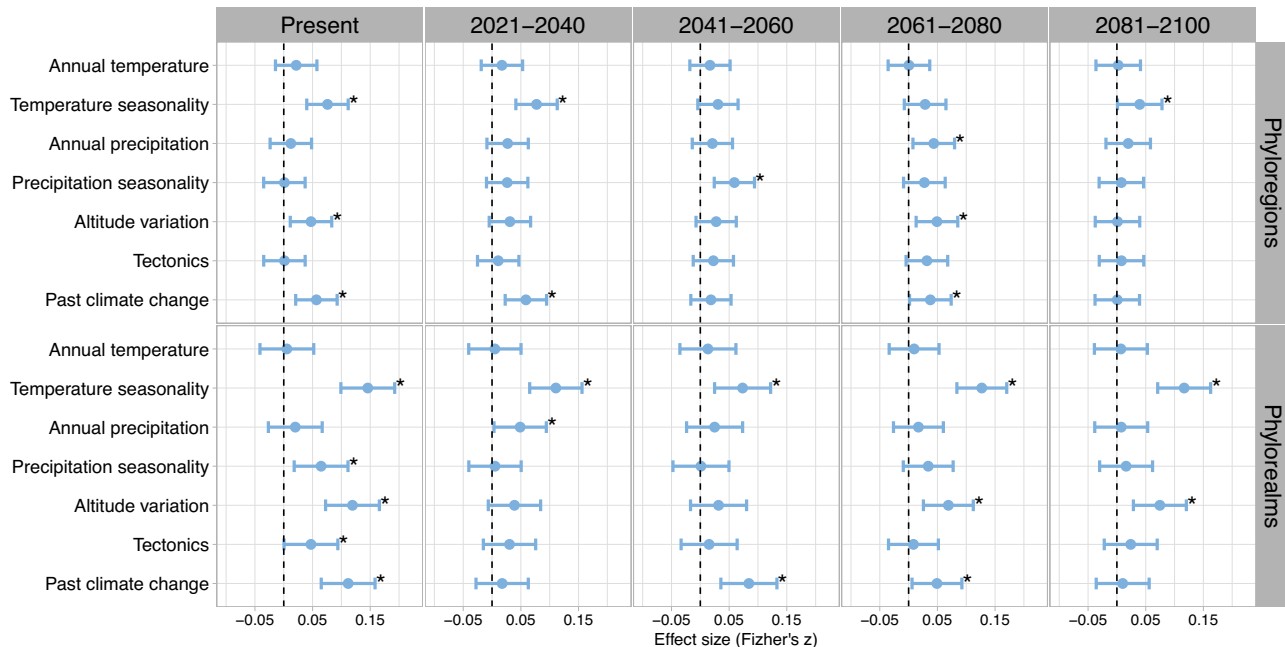

**Fig. 5 | Determinants of plant biogeographic boundaries predicted by a hierarchical generalized model.** The model incorporates predictors hypothesized to determine the positions of biogeographic boundaries including climatic heterogeneity (annual temperature, temperature seasonality, annual precipitation, and precipitation seasonality), orogeny (altitude variation), tectonic movements, velocity of past climate, and spatial autocovariates. Indicated are determinants of all biogeographic boundaries in present-day versus future climate scenarios (2021–2040, 2041–2060, 2061–2080, and 2081–2100) for all region boundaries (top row) and realm boundaries (bottom row). Data are presented as standardized effect sizes calculated using Fisher's z scores with 200 replicates, providing a quantitative representation of the impact of each predictor. The error bars are 95% confidence intervals of z. Statistically significant predictors are marked with asterisks (*), indicating a significance level of $P < 0.05$. Source data are provided as a Source Data file.

other taxonomic groups such as tetrapods[35–37]. Hierarchical generalized linear models[39] show that while present-day region boundaries are driven by areas that have experienced past climate change, and variations in temperature seasonality and altitude (Fig. 5), future region boundaries will be associated with areas that underwent rapid climate change during the Quaternary (T1 and T3) and variations in temperature seasonality in T1 and T4 (Fig. 5). These findings suggest that past climate changes have left lasting legacies that can shape the future redistribution of plant biogeographic region boundaries under anthropogenic change[40–42]. For deeper boundaries (i.e., those separating floristic realms), while temperature seasonality, precipitation seasonality, altitude variation, tectonic separation, and past climate velocity during the Quaternary are important in determining the positions of present-day realm boundaries, only temperature seasonality, altitude variation, and past climate velocity, will remain consistent in shaping future realm boundaries (Fig. 5). Variations in altitude create diverse climatic gradients, impacting temperature, precipitation, and light availability. Species with specific ecological adaptations may find these gradients crucial for survival whereas others may struggle to find suitable habitat especially those at lower elevations. Likewise, variations in temperature seasonality, particularly in mountainous regions, could disrupt phenological cycles (e.g., flowering, or leafing times), pushing species beyond their physiological tolerances. Thus, the physical barrier of mountains and the variation in temperature with altitude will likely continue to be a major factor shaping plant distributions in the coming decades. These findings also suggest that while past movements of tectonic plates, which led to mountain building in some regions, isolation of biotas, or the connectivity of others, have played a role in shaping the biogeographic boundaries we see today, climate change may redistribute the future of plant biogeographic boundaries[43].

Our analysis of global patterns of plant biogeographic regions indicates that future climate change can alter the natural floristic

regions as we know them today, consistent with growing evidence of intensifying biodiversity change[14,15,22,34]. Three general patterns emerge: (i) some floristic regions will remain unchanged while others will see splits and losses due to changes in β-diversity, (ii) deeper boundaries separating floristic realms will suffer more shifts than shallow boundaries, and (iii) boundary shifts are projected to be driven by altitude variation, heterogeneity of temperature seasonality, and past climate velocity.

The biogeography of plants has always been shaped by a complex interaction of biotic and abiotic factors throughout Earth's geological history[9,16]. For example, the Isthmus of Panama may have played a crucial role in the dispersal of tropical and temperate taxa during the Pliocene and Pleistocene ages[44]. The endemic radiation of the flora of the Cape floristic region has been linked to the Benguela upwelling system that brought cold and nutrient-rich waters to the surface along the west coast of South Africa and Namibia during the late Miocene about 10–8 Ma[45,46]. Long-distance and intercontinental dispersal could have been crucial for the formation of the Neotropical forests[47]. Similarly, the Beringian land bridge likely played a role in the dispersal of arctic biota during the Quaternary glaciations[48]. However, our results indicate that as climate change intensifies toward the end of the modern century, the future of these natural and distinctive floristic regions of deep evolutionary origins may be compromised.

Earth's climate has remained relatively stable since the last Ice Age, about 11,500 years ago until ~1800, when global human population first reached 1 billion with the onset of industrialization and enormous expansion in the use of fossil fuels[49,50]. This was followed by a series of abrupt changes that exceed the climate change of the Late Holocene such as the appearance of manufactured materials in sediments, increased greenhouse gas emission, rising sea levels, and unprecedented species invasions and native extirpations[51]. Our study shows that human activities within a relatively short time may surpass the natural geological forces which generated plant biogeographical

regions over millions of years[7,8,16], suggesting that the human enterprise is approaching criticality with significant consequences for Earth system functioning[52]. Our findings reveal that the future of plant communities can be highly dynamic, adding to the need to mitigate climate impacts if the targets of COP 28 for 2030 and 2050 are to be met[53].

## Methods

### Distribution data and species distribution modeling

We explored shifts in plant biogeographic regions under climate change using species distribution modeling under alternative climate change scenarios. Our models were constructed based on the standard protocol for reporting species distribution models using the ODMAP (Overview, Data, Model, Assessment and Prediction) protocol[54] (Supplementary Note 1), along with open-source data and codes for scientific reproducibility (see Data Availability). We used maximum entropy (MaxEnt v.3.4.3)[55] to model plant species distributions. MaxEnt is not computationally expensive[56] and has been shown to outperform other algorithms in modeling species distributions for computational efficiency especially when dealing with a huge number of species spanning hundreds of thousands of species as in this study and is robust for modeling distributions for species with relatively few occurrence records[57]. Predictor variables were downloaded from WorldClim v.2.1 (ref. 58) at a spatial grain resolution of 5-arcmin (equivalent to ~9 km at the equator) for present-day conditions (1970-2000) and four future climate scenarios (T1: 2021-2040, T2: 2041-2060, T3: 2061-2080, and T4: 2081-2100) based on MIROC6 and four Shared Socioeconomic Pathways (SSP 126, 245, 370, and 585). These pathways represent varying levels of climate mitigation, ranging from strong mitigation (SSP126) to moderate (SSP245 and SSP370) and high emissions (SSP585) scenarios[32,33]. We considered 20 predictor variables (Supplementary Table 4) which are hypothesized to be important for plant distributions and diversity in previous studies[59,60]. From these predictor variables, we removed areas corresponding to inland waters, i.e., lakes (using vector polygons from https://naturalearthdata.com). Variance Inflation Factor (VIF) was calculated among predictor pairs to remove highly autocorrelated predictors using the R package usdm version 2.1-6 (ref. 61).

Plant occurrence records were compiled from the Global Biodiversity Information Facility (GBIF, https://doi.org/10.15468/dl.jqqjba, accessed 2 June 2024) using the query term "Tracheophyta". This yielded 454 million records from 14,405 published datasets. However, we previously showed that raw point occurrences of plants suffer from inherent coverage gaps and sampling biases[62] that can hinder their ability to accurately represent global biodiversity patterns[62,63]. To this end, we used a multi-step workflow to address these limitations as follows: (i) Source data: the raw occurrence data used to produce the species' range polygons were obtained from GBIF. (ii) Data cleaning: these records were thoroughly cleaned by matching species names from the GBIF occurrences to those in the World Checklist of Vascular Plants (WCVP) and keeping only verified names from WCVP[64]. At the same time, the point records were refined to capture native distributions by intersecting them with WCVP's native range maps of vascular plants within country borders[64] and retaining points that overlap WCVP's range maps. (iii) Polygon maps: After data cleaning, we converted the point records into polygon maps by modeling with alpha hulls using the R package rangeBuilder v.2.1 (ref. 65). We cropped each species' polygon map to land areas using a basemap from naturalearth (https://naturalearthdata.com). Finally, we systematically sampled these polygon maps to generate 500 points per species for input into the species distribution model (SDM) as in previous studies[66–69] rather than using the raw and biased point occurrences. (iv) Species-specific dispersal rate: We incorporated a partial-dispersal model to prevent erroneous predictions in suitable but unoccupied areas[70,71]. Specifically, we calculated species-specific dispersal rates using a spherical Brownian motion model (SBM)[72,73] implemented with the R package castor v.1.7.10 (ref. 74). Unlike the widely used Brownian Motion models of continuous trait evolution that encode geographic locations in orthogonal space[75,76], the SBM model quantifies the dispersal of a clade over time as a diffusion-like process based on a single diffusion coefficient $D$, while accounting for Earth's spherical geometry[73,74]. The SBM model was fitted using the function *fit_sbm_const* in the R package castor v.1.7.10 (ref. 74). (v) Calibration area: the resulting dispersal rate, defined as the expected dispersal distance traversed by a species in a year (expressed in km/year), was used to define calibration areas (i.e., training areas) for modeling species distributions for each species across different timeframes. This was achieved by buffering the dispersal rates around the alpha hull polygons of each species and intersecting the buffered zones with maps of the terrestrial ecoregions of the world[3] overlapped by the species to predict habitat suitability of each species. This latter step was intended as an additional fine-tuning process to allow us to capture the natural habitats of each species based on their overlap with the ecoregions. (vi) Background points: we generated background points as a function of global plant sampling intensity to account for the biased sampling in the input occurrence records using spatial kernel density estimation and probabilistic sampling of 10,000 background points for each species within their calibration areas. (vii) Species distribution modeling: species distribution modeling was conducted to estimate species distributions based on environmental conditions that correlate with known occurrences, and calibrated to species' realized niche based on the calibration area defined using the species-specific dispersal rates. From the occurrence data as input, we used a 75% random sample for model development, while retaining the remaining 25% sample for model evaluation. For each species, we built models using a combination of hyperparameters in terms of the feature classes and regularization multiplier settings in MaxEnt v.3.4.3 (ref. 54) as follows: linear, threshold, and hinge responses, and tested a set of regularization multiplier values (2, 5, 10, 15, 20) under a 5-folds cross-validation framework[77]. Our models were predicted over each species' occurrences as a function of present-day bioclimatic variables and using these combinations of settings on a continuous scale between 0 and 1 using the *sdm* function in phyloregion v.1.0.9 (ref. 78). We generated five sets of models for each species and took the median to account for uncertainties across different model runs. While ensemble methods that integrate and average predictions from multiple models are valuable, this can be computationally expensive and impractical for large datasets as in our study. With hundreds of thousands of species spanning five time horizons and four climate scenarios, creating ensembles for each would be computationally expensive and time-consuming. Moreover, if the individual models within the ensemble are highly similar, the ensemble may not provide much additional benefit compared to a single model. Therefore, we decided to run each model five times and take the median as a more efficient approach. For future climate scenarios, we modeled plant distributions as a function of present climate variables, and then used these models to predict plant distributions at new values of climate under different future scenarios for T1–T4 and SSP126, SSP245, SSP370, and SSP585. The model prediction consisted of a range map stored in raster format at a 5-arc minute grid cell resolution. The suitability of the models for each species was converted to binary presences by using the 95% quantile of the suitability values extracted from the underlying occurrence records as presence threshold. The final dataset contains range maps for 189,269 species, for the present, and four future time horizons each with four SSPs resulting in a total of 3,217,573 range maps for the analysis of biogeographical regionalization under climate change.

### Biogeographical regionalization

For each climate scenario, we overlaid each species modeled distributions onto equal area grid cells of 100 km × 100 km (Mollweide

global projection system) to convert the predicted distribution data to a community matrix of 189,269 species × 14,810 grid cells using the *polys2comm* function in the R package phyloregion v.1.0.9 (ref. 78). This resulted in a total of 17 different community matrices of 189,269 species × 14,810 grid cells (one for the present condition and four for the four different future time horizons × four SSPs). We then matched the community data to the most updated phylogenetic tree of the world's vascular plants[79]. The phylogenetic tree was generated using the R package V.PhyloMaker2 v.0.1.0 (ref. 80) with function *phylo.maker* based on the expanded megaphylogeny (GBOTB.extended.TPL) of ref. 79 as a backbone and the function *build.nodes.1* in the R package V.PhyloMaker2 v.0.1.0 (ref. 80). Missing taxa from the megaphylogeny were added using V.PhyloMaker2 under scenario "S2" that allows generation of random trees. We generated one tree and used it to analyze compositional turnover (phylogenetic beta diversity) based on Simpson beta diversity index (βsim)[81]. We used the βsim here to measure beta diversity as in previous studies of biogeographical regionalization[4,7,8,12,36,37] because it is insensitive to differences in species richness among sites[82], and therefore provides unbiased estimation of species composition among sites[4]. Simpson beta diversity is expressed as:

$$\beta sim = 1 - \frac{a}{\min(b, c) + a} \qquad (1)$$

where $a$ is the number of species shared between two sites, and $b$ and $c$ are the numbers of species unique to each site. Values of βsim vary from 0 (high similarity in species composition between sites) to 1 (no shared taxa). We additionally tested the robustness of our results by comparing our findings from βsim (Simpson's index) with those obtained from the Sorensen index, which is known to be more sensitive to variations in species richness[82]. We found generally similar results (Supplementary Fig. 6–8), suggesting the reliability of our conclusions based on βsim. Matrices of beta diversity were calculated using the function *phylobeta* in the R package phyloregion v.1.0.9 (ref. 78).

Next, we contrasted the performance of eight different hierarchical clustering algorithms (UPGMA, Single, Complete, ward.D, ward.D2, WPGMA, WPGMC, and UPGMC) on each of the 17 βsim matrices for degree of data distortion using ref. 83's cophenetic correlation coefficient using the *select_linkage* function in phyloregion v.1.0.9 (ref. 78). For all the distance matrices of phylogenetic beta diversity, the unweighted pair group method with arithmetic mean (UPGMA) was identified as the best clustering algorithm (Supplementary Fig. 1) and was used to cluster the distance matrices in downstream analyses.

To determine the optimal number of clusters that best describes the observed βsim matrices, we adapted ref. 12's approach and selected two different thresholds to define floristic regions corresponding to the minimum number of regions that explained 90% of between-cluster βsim (sum of between-cluster βsim/total βsim) and 85% of between-cluster βsim. We refer to the clusters explaining 90% of phylogenetic dissimilarity as 'regions', while clusters explaining 85% of dissimilarity correspond to the 'realms'. The outputs were stored as vector polygon for mapping and visualizations.

## Climate change and plant biogeographic boundaries
We assessed the potential effects of climate change on biogeographic boundaries as the boundaries between floristic regions for which at least one adjacent cell belongs to a different floristic region[35]. This delineation involved four steps. First, for each time horizon and climate scenario, we split the vector polygons based on floristic regions. Second the vector polygons of each floristic region were rasterized and assigned a value of 1 using the *rasterize* function in the R package terra v.1.7-55 (ref. 84). Third, buffers of 200 km were generated around the

cells using the function *buffer* in the R package terra v.1.7-55 (ref. 84). Finally, the rasterized polygons were summed. Raster cells with a cumulative value greater than one correspond to plant biogeographic boundaries.

## Spatial congruence across regionalizations
We measured the degree of spatial association between present versus future regionalizations using a quantitative measure known as v-measure[85]. The v-measure evaluates spatial congruence using two criteria: homogeneity and completeness. A spatial association satisfies homogeneity criteria if all of the regions contain only cells which have a single label. An association satisfies the completeness criteria if all cells having the same label belong to a single region. To this end, we projected each map in the Equal Earth projection system *(+proj=eqearth* code) and assessed congruence between the maps using the V-measure statistic in the R package sabre v.0.4.3 (ref. 85). Spatial association was computed using the function *vmeasure_calc* and setting the option B > 1 so that completeness is weighted more than homogeneity. V-measure scores range from 0 (incongruence) to 1 (indicating perfect similarity).

We additionally evaluated the similarities or differences of present biogeographic boundaries versus future boundaries. This was achieved by comparing the position of terrestrial boundaries across the different regionalizations for both regions and realms. From the spatial raster boundaries described above, we computed the geographic distance of each cell to the nearest boundary using the function *distance* in terra v.1.7-55 (ref. 84). We then conducted a spatially corrected correlation between the position of the present biogeographic boundaries versus future boundaries. The correlations were conducted using a corrected Pearson's correlation for spatial autocorrelation using the function *modified.ttest* in the R package SpatialPack v.0.4 (ref. 86).

## Determinants of biogeographical boundaries
We tested the potential of various biogeographical drivers, such as orographic barriers, tectonic movements, and variations in climate seasonality (temperature and precipitation), in explaining the position of present and future plant biogeographic boundaries. Along these lines, we considered biogeographical drivers hypothesized to affect biogeographical boundaries in previous studies[35–37] and grouped them into four categories: climate heterogeneity, tectonic movements, orographic barriers, and instability of past climate. (1) To assess climate heterogeneity, we obtained four bioclimatic variables known to explain biogeographic patterns in previous global studies[87] including: mean annual temperature, temperature seasonality, mean annual precipitation, and precipitation seasonality, from WorldClim v.2.1 (ref. 58). We acknowledge that our original input maps are modeled estimations based on bioclimatic variables, and using variables solely from WorldClim could introduce some circularity. To address this, we calculated climate heterogeneity for each grid cell as the coefficient of variation between the focal cell and its eight neighbors. This approach assumes that biogeographic boundaries often occur in areas with sharp turnover in climate regimes. Grid cells with higher heterogeneity values indicate they are different from neighboring ones. (2) Tectonic movement was determined by reconstructing the geographic locations of present-day grid cell centroids in timesteps of 1 Ma back to their historical positions 65 Ma. For each grid cell, we calculated the historical distance between a grid cell and eight neighboring cells at each timestep, and then computed the standard deviation of these distances across the last 65 million years, representing the variability of geographical distances between grid cells across time. Our historical distance reconstruction was based on *SETON2012*'s tectonic plate model[88] which reconstructs global plate motion for coastlines and topological plate polygons since the break-up of the supercontinent Pangea 200 Ma. This was implemented using the function *reconstruct*

in the R package rgplates v.0.4.0 (ref. [89]), which is an interface for the GPlates plate reconstruction software[88]. (3) To estimate orographic barriers, we obtained elevation data from WorldClim v.2.1 (ref. [58]) and calculated the mean absolute difference in elevation between a focal grid cell and 8 neighboring cells. (4) We also included past climate velocity during the Quaternary (reconstructed by ref. [90]) at 5-arcmin resolution by extracting values across grid cells. We standardized all predictor variables to have a mean of 0 and variance of 1. All predictor variables showed variance inflation factors less than 2 indicating minimal collinearity.

Next, we used a simultaneous autoregressive spatial (SAR) model with binomial error distribution to assess relationships of the position of plant biogeographic boundaries and the predictors while accounting for spatial autocorrelation. This was achieved by building our models with the response variable being a binomially distributed binary variable which determines whether a given cell is in contact with a biogeographic boundary or not. Along these lines, for each time horizon and climate scenario, we split the vector polygons based on floristic regions, rasterized the positions of boundaries between floristic regions, and generated buffers of 200 km around the boundary cells. The cells directly touching boundary lines were coded as "YES" and the remaining buffered cells not touching the boundary lines were coded "NO". To ensure the best performance of our spatial regression, we incorporated spatial autocorrelation in the error term using neighborhood matrices. The neighborhood matrices were defined based on the diameter of a circle extending from one grid cell centroid to another cell centroid, corresponding to 283 km, being the shortest cell that kept all cells connected in the study area (functions *dnearneigh* and *nb2listw* in the R package spdep v.1.3-1, ref. [91]). We then used the function *hglm* in the R package hglm v.2.2-1 (ref. [92]) to fit hierarchical generalized linear mixed models (HGLMs)[93] with spatially autocorrelated random effects with 200 bootstrap replicates. For each variable, the effect size of the HGLM coefficients was converted to Fisher's *z*, which measures effect size independent of differences in sample sizes[94]. Values of Fisher's *z* were calculated using the t-values from the output of the HGLM models using the *tes* function in the R package compute.es v.0.2-5 (ref. [95]). See 'Data availability' to access the data and analysis codes[96].

**Reporting summary**

Further information on research design is available in the Nature Portfolio Reporting Summary linked to this article.

# Data availability

Plant occurrence records used for the modeling were downloaded from the Global Biodiversity Information Facility (GBIF, https://doi.org/10.15468/dl.jqqjba, accessed 2 June 2024) using the query term "Tracheophyta". From these records, we generated range maps using a combination of biodiversity informatics and species distribution modeling, and the resulting community matrices are archived on Dryad at https://doi.org/10.5061/dryad.xd2547dqc. The phylogenetic tree used for the analysis is a published phylogeny that is already available in public repositories[79]. Specifically, the plant phylogeny was downloaded from Smith & Brown[79]. Source data are provided with this paper.

# Code availability

All scripts and code necessary to repeat the analyses described here have been made available in the new R package phyloregion[78].

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

## Acknowledgements
We thank Stanford University for logistic support. B.H.D. was supported by the US National Science Foundation (awards 2345994 and 2416314).

## Author contributions
The study was conceived and designed by B.H.D. All analyses were carried out by B.H.D. The manuscript was written by B.H.D. The manuscript was revised by B.H.D. and S.M-B. Final approval of the submitted version: B.H.D. and S.M-B.

## Competing interests
The authors declare no competing interests.
