## [Transparent Peer Review file · Nature Communications]

Climate change alters the future of natural floristic regions of deep evolutionary origins

Corresponding Author: Dr Barnabas Daru

Version 0:

Reviewer comments:

Reviewer #1

(Remarks to the Author)

This article used distribution models of 190,942 plant species to define biogeographic boundaries under the current climate and future climates projected by the climate-change scenarios. It also evaluates the changes in the beta diversity of global flora. The topic is relevant for understanding the consequences of climate change on changes in global biota.

I have some comments and questions:

1. My main concern is that the study only models habitat suitability but does not consider species dispersal. It is possible that, in many cases, a new climate can create a suitable habitat that is too far from the current species range, and the species will not be able to disperse to this habitat within a few decades. Therefore, the models of future bioregions and estimates of future beta diversity may be unrealistic. Did you allow overseas migration in your models? Maybe only species reported as naturalized outside their native range (e.g. in the GloNAF database) could be allowed to migrate overseas in the models. Or you could use a model that accounts for limited migration, such as KISSMIG (Nobis and Normand 2014, *Ecography*).
2. I miss a justification for using the Simpson index rather than the Sorensen index to measure beta diversity. Simpson index measures only the turnover component of beta diversity but ignores the nestedness component, which can also be important for defining biogeographical boundaries.
3. The study suggests that phylogenetic beta diversity will be lower in the regions at the end of the century, but isn't it just a consequence of the higher number of regions defined at the end of the century? If there are more regions, more beta diversity occurs among the regions, and consequently, less beta diversity occurs within the regions.
4. Overall, the text is rather tedious to read. It seems that the authors tried to condense much information into a very brief text, and as a consequence, the text is not understandable in places. Also, a lot of technical information, such as the results of statistical tests, is presented in the body of the text, which makes it quite heavy. Furthermore, there are wording issues in several sentences, for example, on pages 108-112: "The areas ... are correlated with residuals": which variable describing the areas is correlated? What does "than" refer to? Do you mean "higher than" or "lower than"? I do not understand this sentence. On pages 134-138, it is also unclear what "than" is related to. Perhaps "more moderate" or "stronger" should be said in the previous part of the sentence? Unfortunately, these issues make it difficult to understand some key findings of this study. The text needs a significant revision to improve clarity and readability.

Minor comments (referred to line numbers):

- 15 - Declines over which period?
- 17 - It is unclear from the Abstract what "deeper boundaries" mean. Actually, it is not defined in the main text either.
- 19 - "areas that experienced ... sharp transitions ... in elevation"? I do not follow.
- 31 - Beta diversity is not a composition of species assemblages. It is a change in this composition.
- 35-36 - Why are Late Triassic and Miocene specifically mentioned?
- 36 - What does it mean "foundational species"?
- 85 - How were the percentage thresholds for regions and realms defined?

67- It should be mentioned how these 190,942 species were selected and from which source. It is explained in detail in Supplementary Information, but basic information should also be given in the main article.

86 - Which units were clustered? It is probably explained on line 127 that it was areas of 100 km², but it is not clear, and it should be clearly explained already here. It only becomes after reading the Supplementary Information, but it should also be clear from the main text of the article.

118-123 - This is a discussion of land-use change (removal and fragmentation of Amazonian forests). I think it is irrelevant here because the models used in this article are only based on climate change, not on land-use change.

150-152 I don't think the parallel with Clements' superorganism is appropriate here. Clements mainly dealt with local plant communities, not with floras of biogeographical regions.

163 Did you really use changes in elevation ("areas that ... experienced ... sharp transitions in ... elevation")? Do you mean changes during orogenesis, or is it a confusing wording?

194-195 More accurately, it was about 11,500 years.

Fig. 2: I am confused by the region codes (numbers) shown in the maps. For example, the Circumboreal region has the code 1 in the present, T1 and T4 maps, but it has the code 2 in the T2 and T3 maps. The Indian-Indochinese region has the code 7 in the graph, but in the maps, it is coded 7, 6, 7, 9, and 8. The same problems occur in other regions. Do I misunderstand something, or are there errors in the figure?

Reviewer #2

(Remarks to the Author)

This is an important and interesting study, which can provide a great contribution to our understanding of how floristic regions will change in response to future climate change. It is generally well-represented too.

However, there are several places that probably need more explanations/information to help readers.

1) Inclusion of the main method section in the main text. In its current version, the authors put all methods in the supplementary materials, which makes it difficult to get the main steps of the analyses, and sometimes the results too.

2) Addition of analysis details. The manuscript references an under-review paper for methodological details, and the authors provided how they extracted the species' occurrence records from GBIF in the summary file, leaving critical information, such as species selection, occurrence data processing, and native status classification, unspecified in the main text. Given the potential discrepancies in species' status across databases like POWO, GIFT, and GloNAF, clarity on how these conflicts were resolved is essential for reproducibility. Even though the number of species in the study is huge, and the conclusions obtained should be robust, this information is necessary in the text.

3) The study fundamentally relies on SDM with the MaxEnt algorithm. I understand it was chosen for computational efficiency given the huge number of species, but I have reservations about using such a method to model distribution for species with few observations. I also noticed that in the authors' previous paper (Daru et al. 2021), an ensemble species distribution model i.e., averaging models over four algorithms (RF, GLM, GBM, and MaxEnt) was used. Therefore, the rationale for not adopting a similar ensemble methodology in this study is unclear and warrants further justification.

4) In Fig. 1A-E, the numbers (1-20) and colors represented different regions, particularly between A and E, however in F-I, they were aligned for the comparisons of changes. In addition, whether the sizes of the regions change for different scenarios? If so, how those different region sizes would affect the comparisons? And whether the authors had tested the significance of the residuals. From the bars, it seems rather few were different.

5) Fig. 4 and relevant results. I found those results a bit hard to understand. As described, the current and future species distributions were generated using SDMs, which linked bioclimatic variables with occurrences and predicted suitable areas based on the links. Thus, species' distribution ranges, or the obtained regions/boundaries are the results of the occurrence-climate models. The authors then used several same and additional variables to test the factors shaping the boundaries. Whether is it a bit cyclic? And I could not understand completely why the wilderness was included. In addition, as shown in L 165-174, the past climate change was the most important and persistent factor, does it indicate that future climate change has rather minimal influence on the obtained changing boundaries?

Minor comments:

1. L65-66, maybe a good place to mention this study is based on occurrence records, rather than regional data, like previous studies.

2. L70, why the SSP126 is a "strong" scenario?

3. L105, Eremean or Tropics-Eremean?

4. L113, could not see these results from Fig. S4.

5. L161, it seems not only "future" boundaries were analyzed here.

6. In the materials and methods, P3, step v, "with known occurrences". The known occurrences referred to the records obtained from the GBIF. P4 L2, what were the "17 communities matrices"? P15, it is a bit hard to understand what the response variable was in the biogeographic boundary regression. From the description, it seems the authors constructed a Y/N variable with cells overlapping with the boundary as 1, and all other grid cells (places in white in Fig. 3) as 0? I am wondering whether the huge amount of 0s would influence the analyses.

7. In Fig.1, perhaps adding a symbol of a phylogenetic tree to "input data" will make it more intuitive.

8. Fig. 4, it seems unclear why the elevation was significant in T2, rather than in other times.

9. In supplementary lines 86-87, which tool was used for the tree construction?

References:

Daru, B.H., Davies, T.J., Willis, C.G., Meineke, E.K., Ronk, A., Zobel, M., et al. (2021). Widespread homogenization of plant communities in the Anthropocene. *Nat. Commun.* 12, 1–10.

Reviewer #3

(Remarks to the Author)

Version 1:

Reviewer comments:

Reviewer #2

(Remarks to the Author)

I am delighted with the revisions of Manuscript #NCOMMS-24-04378A, Climate change alters the future of natural floristic regions of deep evolutionary origins. The authors have admirably addressed my concerns, impressively reanalysing such an extensive dataset within such a short period. The MS is now both highly readable and informative.

Below are some minor comments:

L138-139, not sure I understand this sentence, particularly about the sea.

L148 & L 159, Fig. 3 should be Fig. 4.

L178-182, I did not see a significant correlation between tectonic separation and present-day realm boundaries, in contrast, tectonics seems positively (although weakly) linked to realm boundaries in the time horizon of 2081-2100.

L187 and some other relevant places, what did the "abrupt transitions in climate seasonality" refer to?

L187-191, did the authors mean that the temperature seasonality only occurs in mountainous areas?

L191-193, many other studies (e.g., Leprieur et al., 2011; Svenning et al., 2011; Xu et al., 2023) have also revealed the importance of paleoclimatic legacy in shaping the spatial turnover of biodiversity, thus, it is recommended to add some references to strengthen this point.

References:

F. Leprieur, P. A. Tedesco, B. Hugueny, O. Beauchard, H. H. Durr, S. Brosse, T. Oberdorff, Partitioning global patterns of freshwater fish beta diversity reveals contrasting signatures of past climate changes. *Ecol. Lett.* 14, 325–334 (2011).

J. C. Svenning, C. Flojgaard, A. Baselga, Climate, history and neutrality as drivers of mammal beta diversity in Europe: Insights from multiscale deconstruction. *J. Anim. Ecol.* 80, 393–402 (2011).

Xu W-B, Guo W-Y, Serra-Diaz J, Schrod F, Eiserhardt W, Enquist B, Maitner B, Merow C, Violle C, Anand M et al. 2023. Global beta-diversity of angiosperm trees is shaped by quaternary climate change. *Sci. Adv.* 9: eadd8553.

L244, should be Table S4, and please check the citation numbers in the table.

Ref. 88 and 89, are they the same?

Reviewer #3

(Remarks to the Author)

REVIEWER COMMENTS

Reviewer #1 (Remarks to the Author):

This article used distribution models of 190,942 plant species to define biogeographic boundaries under the current climate and future climates projected by the climate-change scenarios. It also evaluates the changes in the beta diversity of global flora. The topic is relevant for understanding the consequences of climate change on changes in global biota.

I have some comments and questions:

1. My main concern is that the study only models habitat suitability but does not consider species dispersal. It is possible that, in many cases, a new climate can create a suitable habitat that is too far from the current species range, and the species will not be able to disperse to this habitat within a few decades. Therefore, the models of future bioregions and estimates of future beta diversity may be unrealistic. Did you allow overseas migration in your models? Maybe only species reported as naturalized outside their native range (e.g. in the GloNAF database) could be allowed to migrate overseas in the models. Or you could use a model that accounts for limited migration, such as KISSMIG (Nobis and Normand 2014, Ecography).

RESPONSE 1.1: We thank the reviewer for this comment. We acknowledge that our initial description of the methodology may not have been clear regarding how we considered dispersal limitations. We now clarify that our study primarily focuses on how anthropogenic climate change impacts biogeographic regions defined by the native ranges of vascular plants. To this end, we specifically removed non-native distributions prior to analysis including those defined in the GloNAF database to isolate the effect of climate change on native plant communities within their evolutionary origins. To address dispersal limitations and predict realistic future distributions, we used a novel approach that integrates species-specific dispersal capabilities in defining the calibration area (i.e., training area). This approach restricts predictions to the realized niche (accessible and suitable areas) rather than the fundamental niche (all potentially suitable locations, irrespective of accessibility). We used a Spherical Brownian Motion model to calculate dispersal rate for each species over time. This model treats dispersal as a diffusion-like process across the Earth's geometry, considering a single diffusion coefficient (D). This approach allows us to estimate dispersal rates (expressed in km/year) specific to each species. We then used the resulting dispersal rate to define calibration areas (i.e., training areas) for modelling species distributions for each species across different timeframes. This was achieved by buffering the dispersal rates around the alpha hull polygons of each species and intersecting the buffered zones with maps of the terrestrial ecoregions of the world overlapped by the species to predict habitat suitability of each species. This latter step was intended as an additional fine-tuning process to allow us to capture the natural habitats of each species based on their overlap with the ecoregions. We have provided this information in the revised manuscript in Lines 254–282 as follows:

“(i) Source data: The raw occurrence data used to produce the species’ range polygons were obtained from GBIF (<https://doi.org/10.15468/dl.vgvc3z>). (ii) Data cleaning: These records were thoroughly cleaned by matching species names from the GBIF occurrences to those in the World Checklist of Vascular Plants (WCVF) and keeping only verified names from WCVF⁶⁴. At the same time, the point records were refined to capture native distributions by intersecting them with WCVF’s native range maps of vascular plants within country borders⁶⁴ and retaining points that overlap WCVF’s range maps. (iii) Polygon maps: After data cleaning, we converted the point records into polygon maps by modeling with alpha hulls using the R package rangeBuilder v.2.1

(ref.⁶⁵). We cropped each species' polygon map to land areas using a basemap from *naturalearth* (<https://naturalearthdata.com>). Finally, we systematically sampled these polygon maps to generate 500 points per species for input into the species distribution model (SDM) as in previous studies^{66–69} rather than using the raw and biased point occurrences. (iv) Species-specific dispersal rate: We incorporated a partial-dispersal model to prevent erroneous predictions in suitable but unoccupied areas^{70,71}. Specifically, we calculated species-specific dispersal rates using a spherical Brownian motion model (SBM)^{72,73} implemented with the R package *castor* v.1.7.10 (ref.⁷⁴). Unlike the widely used Brownian Motion models of continuous trait evolution that encode geographic locations in orthogonal space^{75,76}, the SBM model quantifies the dispersal of a clade over time as a diffusion-like process based on a single diffusion coefficient D , while accounting for Earth's spherical geometry^{73,74}. The SBM model was fitted using the function `fit_sbm_const` in the R package *castor* v.1.7.10 (ref.⁷⁴). (v) Calibration area: The resulting dispersal rate, defined as the expected dispersal distance traversed by a species in a year (expressed in km/year), was used to define calibration areas (i.e., training areas) for modelling species distributions for each species across different timeframes. This was achieved by buffering the dispersal rates around the alpha hull polygons of each species and intersecting the buffered zones with maps of the terrestrial ecoregions of the world³ overlapped by the species to predict habitat suitability of each species. This latter step was intended as an additional fine-tuning process to allow us capture the natural habitats of each species based on their overlap with the ecoregions.”

2. I miss a justification for using the Simpson index rather than the Sorensen index to measure beta diversity. Simpson index measures only the turnover component of beta diversity but ignores the nestedness component, which can also be important for defining biogeographical boundaries.

RESPONSE 1.2: We thank the reviewer for this comment, and we acknowledge that Simpson index focuses on turnover, while the Sorensen index can capture nestedness. Our decision to use Simpson index was driven by the nature of our underlying data. Because our analyses rely on modelled estimations aggregated as presence/absence data, we lacked the abundance information required for Sorensen index. Accordingly, we clarify that the Simpson index is advantageous for presence/absence data, as used in our study. We highlight that previous studies have found Simpson index effective for biogeographic regionalization with similar data types. We additionally tested the robustness of our results to the choice of beta diversity metric by re-running our analysis using Sorensen index and re-delineating plant biogeographic regions produced in the main analysis. We found that changes in plant biogeographic regions remained generally similar (Fig. S6 to S8 in the revised manuscript), reinforcing our conclusions that climate change can alter plant biogeographic regions. This information is now provided in the revised manuscript in Lines 327–340 as follows:

“We ... analyzed compositional turnover (phylogenetic beta diversity) based on Simpson beta diversity index (β_{sim})⁸¹. We used the β_{sim} here to measure beta diversity as in previous studies of biogeographical regionalization^{4,7,8,33,36,37} because it is insensitive to differences in species richness among sites⁸², and therefore provides unbiased estimation of species composition among sites⁴. Simpson beta diversity is expressed as:

$$\beta_{sim} = 1 - \frac{a}{\min(b, c) + a}$$

where a is the number of species shared between two sites, and b and c are the numbers of species unique to each site. Values of β_{sim} vary from 0 (high similarity in species composition between sites) to 1 (no shared taxa). We additionally tested the robustness of our results by comparing our findings from β_{sim} (Simpson's index) with

those obtained from the Sorensen index, which is known to be more sensitive to variations in species richness⁸². We found generally similar results (Fig. S6 to S8), suggesting the reliability of our conclusions based on β sim.”

3. The study suggests that phylogenetic beta diversity will be lower in the regions at the end of the century, but isn't it just a consequence of the higher number of regions defined at the end of the century? If there are more regions, more beta diversity occurs among the regions, and consequently, less beta diversity occurs within the regions.

RESPONSE 1.3: The reviewer raised a valid point about possible effect of the number of grid cells on phylogenetic beta diversity. We acknowledge that simply increasing the number of regions could lead to a decrease in within-region beta diversity. To address this concern, we re-analyzed the data to allow a more robust comparison. We used a data-driven approach to determine the optimal number of clusters for each time period and climate scenario, i.e., minimum number of regions that explained 90% of between-cluster β sim (sum of between-cluster β sim/total β sim) and 85% of between-cluster β sim. This means that we did not predefine the number of clusters but instead let the data itself dictate the most suitable number based on its inherent patterns. This approach resulted in a range of optimal number of clusters across scenarios, varying from 21 regions in the present day (T1: 2021-2040) to 25 by 2100. Importantly, these numbers do not directly correspond across different time periods and climate scenarios, indicating that the identified phyloregions are not simply a one-to-one match over time. Thus, to allow a fair comparison across time and climate scenarios, we instead measured shifts in phylogenetic beta diversity (phylobeta diversity) within grid cells relative to the present-day floristic regions. This approach makes it straightforward to directly compare changes in phylogenetic beta diversity within regions across different scenarios and time periods. This information is now provided in the revised manuscript in Lines 94–102 and 116–126 as follows (boldfaced and underlined for emphasis):

Lines 94–102

*“Using the minimum number of clusters explaining 85% (corresponding to floristic realms) and 90% (regions) of Simpson’s phylogenetic dissimilarity based on the unweighted pair group method with arithmetic mean (Fig. S1) as in previous studies^{4,7-9,12}, we identified **19** distinct clusters of 100 km × 100 km grid cells which we define as floristic regions nested within 10 highly distinctive clusters (floristic realms) in the present-day (Fig. 2). The number of clusters for the present-day floristic regions were consistent with future floristic regions for some climate scenarios and different for others, ranging from **21** regions in timeframe T1 (2021-2040) to **25** by 2100 (Fig. S2). We used these thresholds to map plant biogeographic regions for future distributions (Fig. 2).”*

Results (Lines 116–126):

“To assess changes in the composition of biogeographic regions, we measured shifts in phylogenetic beta diversity (“phylobeta diversity”)—the standard metric commonly used to delineate biogeographic regions^{4,12}—across future projections in grid cells relative to present-day floristic regions. We found that future phylobeta diversity is projected to be relatively lower when considering phylobeta diversity within present-day biogeographic regions, with a global mean reduction in Cohen’s d effect size of -0.0058 in T1 to -0.06 in T4 (2081-2100) (Fig. 3, Table S1). The highest spread of negative and significant ($P < 0.05$) Cohen’s d effect size is projected in the Circumboreal, North American Atlantic, Madagascar, Tropics-Eremean and Neozylanic floristic regions (Fig. 3), and this will intensify toward the end of the century in T4 (2081-2100). The projected decline of phylobeta diversity within floristic regions supports our hypothesis that climate change can homogenize plant biogeographic regions.”

4. Overall, the text is rather tedious to read. It seems that the authors tried to condense much information into a very brief text, and as a consequence, the text is not understandable in places. Also, a lot of technical information, such as the results of statistical tests, is presented in the body of the text, which makes it quite heavy. Furthermore, there are wording issues in several sentences, for example, on pages 108-112: "The areas ... are correlated with residuals": which variable describing the areas is correlated? What does "than" refer to? Do you mean "higher than" or "lower than"? I do not understand this sentence. On pages 134-138, it is also unclear what "than" is related to. Perhaps "more moderate" or "stronger" should be said in the previous part of the sentence? Unfortunately, these issues make it difficult to understand some key findings of this study. The text needs a significant revision to improve clarity and readability.

RESPONSE 1.4: We apologize that our previous wording of the results was not clear. In this revision, we have streamlined the reporting of the results to improve clarity and readability. We have also used simpler language to reflect that present-day biogeographic regions and boundaries show lower correlation compared to the correlation among future regions. This information is provided in the revised manuscript as follows:

Floristic regions (Lines 102–105):

“Our delineated plant biogeographic regions in the present-day show moderate spatial correlation with future biogeographic regions, and slightly lower on average compared to the correlation among future biogeographic regions for most time periods (Fig. S3).”

Floristic boundaries (Lines 149–157):

“These differences are further reflected in the spatial correlation of present-day and future boundaries where present-day region boundaries show a weaker correlation with projected future region boundaries combined across climate scenarios compared to the correlation among individual scenarios of future boundaries for periods T2-T4. For floristic realms, our projected realm boundaries showed pronounced incongruence to boundaries of present-day floristic realms compared to those observed for all region boundaries. Present-day realm boundaries are weakly correlated with future realm boundaries compared to the correlation among future realm boundaries.”

Minor comments (referred to line numbers):

15 - Declines over which period?

RESPONSE 1.5: We have revised wording to reflect that the declines in phylobeta diversity is projected for years 2040 to 2100 as follows:

“Our analysis reveals declines in phylogenetic beta diversity for years 2040 to 2100, leading to a future homogenization of biogeographic regions.”

17 - It is unclear from the Abstract what "deeper boundaries" mean. Actually, it is not defined in the main text either.

RESPONSE 1.6: We have revised wording for deeper boundaries to refer to boundaries separating biogeographic realms i.e., highly dissimilar species assemblages as follows:

“While some biogeographic boundaries will persist, climate change will alter boundaries separating biogeographic realms.”

19 - "areas that experienced ... sharp transitions ... in elevation"? I do not follow.

RESPONSE 1.7: Following significant revision of the manuscript, we have reworded this section to improve clarity as follows:

“Such boundary alterations will be determined by variations in past climate change during the Quaternary, altitude variation and heterogeneity of temperature seasonality.”

31 - Beta diversity is not a composition of species assemblages. It is a change in this composition.

RESPONSE 1.8: Thank you. We have revised wording to “*change in species composition*” as suggested.

35-36 - Why are Late Triassic and Miocene specifically mentioned?

RESPONSE 1.9: We have revised this sentence to indicate a broader point about vulnerability of long-established biogeographic regions to climate change follows:

“Concomitantly, the pronounced impact of human-induced global change on biodiversity^{13–15} poses a threat to biogeographic regions which have their origins in deep evolutionary times^{7,16}.”

36 - What does it mean “foundational species”?

RESPONSE 1.10: We have revised wording to indicate plants as primary producers as follows:

“The effects may be severe when they affect primary producers like vascular plants, because the extirpation or change in plant species assemblages can simplify and disrupt ecosystem functioning^{17,18}.”

85 - How were the percentage thresholds for regions and realms defined?

RESPONSE 1.11: In this revision, we clarify that our definition of the optimal number of clusters for defining biogeographic regions and realms followed established thresholds from previous studies of biogeographical regionalization. Following this approach, biogeographic regions were defined as the minimum number of clusters required to explain 85% of phylogenetic dissimilarity whereas biogeographic realms were defined as the minimum number of clusters required to explain 90% of phylogenetic dissimilarity. This information is now provided in the revised manuscript in Lines 94–99 as follows (boldfaced and underlined for emphasis):

*“Using the minimum number of clusters explaining 85% (corresponding to floristic realms) and 90% (regions) of Simpson’s phylogenetic dissimilarity based on the unweighted pair group method with arithmetic mean (Fig. S1) **as in previous studies^{4,7–9,12}**, we identified 19 distinct clusters which we define as floristic regions nested within 10 highly distinctive clusters (floristic realms) in the present-day.”*

67- It should be mentioned how these 190,942 species were selected and from which source. It is explained in detail in Supplementary Information, but basic information should also be given in the main article.

RESPONSE 1.12: The reviewer raised a great point. We have migrated the Methodology from the supplementary information to the main article to provide easy access to important details about our workflow. Accordingly, we revised wording to show that our analysis incorporated 402 million occurrence records for vascular plants resulting in individual species-level range maps for 189,269 species. This selection of 189,269 species reflects those with successfully modelled range maps that are consistent across time horizons and climate scenarios (SSPs) to allow a

one-to-one comparison across climate scenarios. This information is now provided in the revised manuscript as follows:

“Specifically, we used species distribution models to analyze 402 million occurrence records for vascular plants resulting in individual species-level native range maps for 189,269 species under present and future climatic projections throughout the twenty-first century. The selection of the 189,269 species reflects those with successfully modeled distributions that are consistent across different time horizons and climate scenarios.”

86 - Which units were clustered? It is probably explained on line 127 that it was areas of 100 km², but it is not clear, and it should be clearly explained already here. It only becomes after reading the Supplementary Information, but it should also be clear from the main text of the article.

RESPONSE 1.13: We have revised wording to improve clarity and to reflect that our clusters were based on plant species assemblages across 100 km × 100 km grid cells. This information is now provided in the revised manuscript as follows:

“Using the minimum number of clusters explaining 85% (corresponding to floristic realms) and 90% (regions) of Simpson’s phylogenetic dissimilarity based on the unweighted pair group method with arithmetic mean (Fig. S1) as in previous studies^{4,7-9,12}, we identified 19 distinct clusters of 100 km × 100 km grid cells which we define as floristic regions nested within 10 highly distinctive clusters (floristic realms) in the present-day (Fig. 2).”

118-123 - This is a discussion of land-use change (removal and fragmentation of Amazonian forests). I think it is irrelevant here because the models used in this article are only based on climate change, not on land-use change.

RESPONSE 1.14: We agree with this comment. Following significant revision of the manuscript to improve clarity and remove redundancy, we have removed discussion about land-use change without losing meaning in the key findings that climate change can alter plant biogeographic regions.

150-152 I don't think the parallel with Clements' superorganism is appropriate here. Clements mainly dealt with local plant communities, not with floras of biogeographical regions.

RESPONSE 1.15: Thank you. Accordingly, we have removed discussion of the comparison to Clement's superorganism to improve clarity.

163 Did you really use changes in elevation ("areas that ... experienced ... sharp transitions in ... elevation")? Do you mean changes during orogenesis, or is it a confusing wording?

RESPONSE 1.16: We apologize that our previous wording was not clear. We now revised wording in the methodology to clarify our intention to estimate the role of contemporary orographic barriers rather than changes during orogenesis. This was achieved by using elevation data from WorldClim and calculating the mean absolute difference in elevation between a focal grid cell and eight neighboring cells. This information is now provided in the revised manuscript in Lines 413–415 as follows:

“To estimate orographic barriers, we obtained elevation data from WorldClim v.2.1 (ref.⁵⁸) and calculated the mean absolute difference in elevation between a focal grid cell and 8 neighboring cells.”

194-195 More accurately, it was about 11,500 years.

RESPONSE 1.17: We thank the reviewer for this suggestion. We have revised wording to reflect 11,500 years.

Fig. 2: I am confused by the region codes (numbers) shown in the maps. For example, the Circumboreal region has the code 1 in the present, T1 and T4 maps, but it has the code 2 in the T2 and T3 maps. The Indian-Indochinese region has the code 7 in the graph, but in the maps, it is coded 7, 6, 7, 9, and 8. The same problems occur in other regions. Do I misunderstand something, or are there errors in the figure?

RESPONSE 1.18: We apologize for the confusion caused by the inconsistency in the region codes (numbers) in our previous submission across the maps in Figure 2. In this revised manuscript, we now clarify that the numbers are for **visual reference within each map** and **do not represent a unique and persistent identifier** for a specific biogeographic region across different time periods. Biogeographic regions can shift and change over time, so a one-to-one correspondence between regions and numbers across different time horizons or climate scenarios would not be accurate. For instance, in the revised Fig. 2 below, the numbers simply help to distinguish clusters within each individual map and the corresponding NMDS plot on the left. They are not meant to imply a direct connection between regions across the timeframes. This information is provided in the revised manuscript in Lines 685–701 (underlined and boldfaced for emphasis):

Fig. 2. Changes in vascular plant biogeographic regions under current and future climate scenarios in geographic space and in NMDS ordination space. Top, global floristic regions of plants in the present-day delineated by clustering modelled range maps for 189,269 vascular plant species with phylogenetic information and applying pairwise Simpson's β -diversity between 100 km \times 100 km grid cells. Bottom, future floristic regions. Future species distributions were predicted by first modeling current plant distributions as a function of current environmental variables and then using this model to predict future plant distributions at new values of climate under different future scenarios and then using that to generate floristic regions for T1: 2021-2040 (see supplementary information for other time periods T2: 2041-2060, T3: 2061-2080, and T4: 2081-2100). The colors in the map and NMDS plots are identical and indicate levels of differentiation of the flora in different floristic regions such that floristic regions with similar colors have similar clades and those with different colors differ in the plant clades they enclose. Black lines separate floristic realms, while white lines separate floristic regions. The numbers in the map and NMDS plots are arbitrary and meant for visual reference to identify clusters for each time period and do not represent a one-to-one match across time periods. The maps are in the equal-area World Mollweide projection.

Reviewer #2 (Remarks to the Author):

This is an important and interesting study, which can provide a great contribution to our understanding of how floristic regions will change in response to future climate change. It is generally well-represented too.

However, there are several places that probably need more explanations/information to help readers.

RESPONSE 2.1: We thank the reviewer for the positive remark on our paper. In this revision, we have clarified all ambiguity and provided more explanations and information to flesh out the novelty of our key finding of how climate change can alter biogeographical patterns.

1) Inclusion of the main method section in the main text. In its current version, the authors put all methods in the supplementary materials, which makes it difficult to get the main steps of the analyses, and sometimes the results too.

RESPONSE 2.2: In response to Reviewer #1's similar comment above (RESPONSE 1.12), we have migrated the Methods section to the main text to make it easier to follow our integrative workflow.

2) Addition of analysis details. The manuscript references an under-review paper for methodological details, and the authors provided how they extracted the species' occurrence records from GBIF in the summary file, leaving critical information, such as species selection, occurrence data processing, and native status classification, unspecified in the main text. Given the potential discrepancies in species' status across databases like POWO, GIFT, and GloNAF, clarity on how these conflicts were resolved is essential for reproducibility. Even though the number of species in the study is huge, and the conclusions obtained should be robust, this information is necessary in the text.

RESPONSE 2.3: We thank the reviewer for raising this comment. In this revision, we have provided full methodological details on the distribution data and species distribution modeling so that the manuscript stands alone without making reference to a separate manuscript under review. Specifically, our revision provides information on the algorithm used for modeling species distributions, the predictor variables for both current and future species distributions. Plant occurrence records used for the modeling were compiled from the Global Biodiversity Information Facility. These records were processed in seven steps to overcome sampling biases in the input data:

- (i) Source data: The raw occurrence data used to produce the species' range polygons were obtained from GBIF.
- (ii) Data cleaning: These records were thoroughly cleaned by matching species names from the GBIF occurrences to those in the World Checklist of Vascular Plants (WCVP) and keeping only verified names from WCVP. At the same time, the point records were refined to capture native distributions by intersecting them with WCVP's native range maps of vascular plants within country borders and retaining points that overlap WCVP's range maps.
- (iii) Polygon maps: We converted the point records into polygon maps by modeling with alpha hulls. Finally, we systematically sampled these polygon maps to generate 500 points per species for input into the species distribution model (SDM) following previous studies rather than using the raw and biased point occurrences.
- (iv) Species-specific dispersal rate: We incorporated a partial-dispersal model to prevent erroneous predictions in suitable but unoccupied areas. Specifically, we calculated species-specific dispersal rates using a spherical Brownian motion model (SBM).

- The SBM model quantifies the dispersal of a clade over time as a diffusion-like process based on a single diffusion coefficient D , while accounting for Earth's spherical geometry.
- (v) Calibration area: The resulting dispersal rate, defined as the expected dispersal distance traversed by a species in a year (expressed in km/year), was used to define calibration areas (i.e., training areas) for modelling species distributions for each species across different timeframes. This was achieved by buffering the dispersal rates around the alpha hull polygons of each species and intersecting the buffered zones with maps of the terrestrial ecoregions of the world overlapped by the species to predict habitat suitability of each species. This latter step was intended as an additional fine-tuning process to allow us capture the natural habitats of each species based on their overlap with the ecoregions.
 - (vi) Background points: We generated background points as a function of global plant sampling intensity to account for the biased sampling in the input occurrence records using spatial kernel density estimation and probabilistic sampling of 10,000 background points for each species within their calibration areas.
 - (vii) Species distribution modeling: Species distribution modeling was conducted to estimate species distributions based on environmental conditions that correlate with known occurrences, and calibrated to species' realized niche based on the calibration area defined using the species-specific dispersal rates. Our models were predicted over each species' occurrences as a function of present-day bioclimatic variables and using various combinations of settings on a continuous scale between 0 and 1. For future climate scenarios, we modelled plant distributions as a function of present climate variables, and then used these models to predict plant distributions at new values of climate under different future scenarios for T1–T4 and SSP126, SSP245, SSP370, and SSP585. The model prediction consisted of a range map stored in raster format at a 5-arc minute grid cell resolution. The suitability of the models for each species was converted to binary presences by using the 95% quantile of the suitability values extracted from the underlying occurrences records as presence threshold. The final dataset contains range maps for 189,269 species, for the present, and four future time horizons each with four SSPs resulting in a total of 3,217,573 range maps for the analysis of biogeographical regionalization under climate change.

This information is now clearly provided in the revised manuscript in Lines 249–313 so that the paper stands alone without referring to an external manuscript for such details.

3) The study fundamentally relies on SDM with the MaxEnt algorithm. I understand it was chosen for computational efficiency given the huge number of species, but I have reservations about using such a method to model distribution for species with few observations. I also noticed that in the authors' previous paper (Daru et al. 2021), an ensemble species distribution model i.e., averaging models over four algorithms (RF, GLM, GBM, and MaxEnt) was used. Therefore, the rationale for not adopting a similar ensemble methodology in this study is unclear and warrants further justification.

RESPONSE 2.4: We thank the reviewer for raising this point. As the reviewer noted, our reasoning for using MaxEnt was for computational efficiency. However, to account for uncertainties across different model runs, we generated five sets of models for each species and took the median. While ensemble methods that integrate and average predictions from multiple models are valuable, this can be computationally expensive and impractical for large datasets as in our study. With hundreds of thousands of species spanning five time horizons and four climate scenarios for final dataset of 3,217,573 range maps, creating ensembles for

each would be computationally expensive and time-consuming. Moreover, if the individual models within the ensemble are highly similar, the ensemble may not provide much additional benefit compared to a single model. Therefore, we decided to run each MaxEnt model five times and take the median as a more efficient approach. This information is provided in the revised manuscript in Lines 296–313 as follows:

“We generated five sets of models for each species and took the median to account for uncertainties across different model runs. While ensemble methods that integrate and average predictions from multiple models are valuable, this can be computationally expensive and impractical for large datasets as in our study. With hundreds of thousands of species spanning five time horizons and four climate scenarios, creating ensembles for each would be computationally expensive and time-consuming. Moreover, if the individual models within the ensemble are highly similar, the ensemble may not provide much additional benefit compared to a single model. Therefore, we decided to run each model five times and take the median as a more efficient approach. For future climate scenarios, we modelled plant distributions as a function of present climate variables, and then used these models to predict plant distributions at new values of climate under different future scenarios for T1–T4 and SSP126, SSP245, SSP370, and SSP585. The model prediction consisted of a range map stored in raster format at a 5-arc minute grid cell resolution. The suitability of the models for each species was converted to binary presences by using the 95% quantile of the suitability values extracted from the underlying occurrences records as presence threshold. The final dataset contains range maps for 189,269 species, for the present, and four future time horizons each with four SSPs resulting in a total of 3,217,573 range maps for the analysis of biogeographical regionalization under climate change.”

4) In Fig. 1A-E, the numbers (1-20) and colors represented different regions, particularly between A and E, however in F-I, they were aligned for the comparisons of changes. In addition, whether the sizes of the regions change for different scenarios? If so, how those different region sizes would affect the comparisons? And whether the authors had tested the significance of the residuals. From the bars, it seems rather few were different.

RESPONSE 2.5: We apologize for that our previous description of the figure might have been unclear. In response to reviewer 1’s similar comment above (RESPONSE 1.18), we have made substantial revision to this figure to improve clarity. We now clarify that the numbers in the maps are for **visual reference within each map** and **do not represent a unique and persistent identifier** for a specific biogeographic region across different time periods. Biogeographic regions can shift and change over time, so a one-to-one correspondence between regions and numbers across different time horizons or climate scenarios would not be accurate. Likewise, the comparison of previous Fig. 2F-I has now been revised as new Fig. 3 below, which quantifies changes in phylogenetic beta diversity under climate change relative to present floristic regions. This means that the magnitude of change in β -diversity across spatial and temporal scales was assessed by comparing the **grid-cell compositional dissimilarity** for delineating present vs future floristic regions when considering β -diversity within present-day floristic regions using t -test followed by Cohen’s d with 1000 bootstrap replicates to estimate effect size. We found that future phylobeta diversity is projected to be relatively lower when considering phylobeta diversity within present-day biogeographic regions, with a global mean reduction in Cohen’s d effect size of -0.0058 in T1 to -0.06 in T4 (2081-2100). The highest spread of negative and significant ($P < 0.05$) Cohen’s d effect size is projected in the Circumboreal, North American Atlantic, Madagascan, Eremean and Neozylanic floristic regions, and this will intensify toward the end of the century in T4 (2081-2100). The projected

decline of phylobeta diversity within floristic regions supports our hypothesis that climate change can homogenize plant biogeographic regions.

Fig. 3. Changes in phylogenetic beta diversity under climate change relative to present floristic regions. The magnitude of change in β -diversity across spatial and temporal scales was assessed by comparing the grid-cell compositional dissimilarity for delineating present vs future floristic regions when considering β -diversity within present-day floristic regions using t -test followed by Cohen's d with 1000 bootstrap replicates to estimate effect size. Values of Cohen's d range from 0 (no effect) to +1 or -1 (large effect), with positive values indicating differentiation, whereas negative values indicate homogenization. The error bars indicate 95% confidence intervals, and the statistical significance of the t -test are indicated with asterisks ($P < 0.01$).

Regarding whether the sizes of the regions change for different scenarios, we here clarify that because biogeographic regions can shift and change over time, a one-to-one correspondence between regions and numbers across different time horizons or climate scenarios would not be accurate. Instead, we measured spatial congruence across regionalizations using a quantitative measure known as v -measure which evaluates spatial congruence across categorical maps. This information is now provided in the revised manuscript in Lines 368–387 as follows:

“Spatial congruence across regionalizations

We measured the degree of spatial association between present versus future regionalizations using a quantitative measure known as v -measure⁸⁵. The v -measure evaluates spatial congruence using two criteria: homogeneity and completeness. A spatial association satisfies homogeneity criteria if all of regions contain only cells which have a single label. An association satisfies the completeness criteria if all cells having the same label belong to a single region. To this end, we projected each map in the

Equal Earth projection system (+proj=eqearth code) and assessed congruence between the maps using the V-measure statistic in the R package sabre v.0.4.3 (ref.⁸⁵). Spatial association was computed using the function vmeasure_calc and setting the option B > 1 so that completeness is weighted more than homogeneity. V-measure scores range from 0 (incongruence) to 1 (indicating perfect similarity).

We additionally evaluated the similarities or differences of present biogeographic boundaries versus future boundaries. This was achieved by comparing the position of terrestrial boundaries across the different regionalizations for both regions and realms. From the spatial raster boundaries described above, we computed the geographic distance of each cell to the nearest boundary using the function distance in terra v.1.7-55 (ref.⁸⁴). We then conducted a spatially corrected correlation between the position of the present biogeographic boundaries versus future boundaries. The correlations were conducted using a corrected Pearson's correlation for spatial autocorrelation using the function modified.ttest in the R package SpatialPack v.0.4 (ref.⁸⁶)."

5) Fig. 4 and relevant results. I found those results a bit hard to understand. As described, the current and future species distributions were generated using SDMs, which linked bioclimatic variables with occurrences and predicted suitable areas based on the links. Thus, species' distribution ranges, or the obtained regions/boundaries are the results of the occurrence-climate models. The authors then used several same and additional variables to test the factors shaping the boundaries. Whether is it a bit cyclic? And I could not understand completely why the wilderness was included. In addition, as shown in L 165-174, the past climate change was the most important and persistent factor, does it indicate that future climate change has rather minimal influence on the obtained changing boundaries?

RESPONSE 2.6: The reviewer raised an important point. We have accordingly revised analysis of the determinants of biogeographic regions by considering biogeographical drivers hypothesized to affect biogeographical boundaries in previous studies³⁵⁻³⁷ and grouped them into four categories: climate heterogeneity, tectonic movements, orographic barriers, and instability of past climate. (1) To assess climate heterogeneity, we obtained four bioclimatic variables known to explain biogeographic patterns in previous global studies including: mean annual temperature, temperature seasonality, mean annual precipitation and precipitation seasonality, from WorldClim v.2.1. **We acknowledge that our original input maps are modeled estimations based on bioclimatic variables, and using variables solely from WorldClim could introduce some circularity.** To address this, we calculated climate heterogeneity for each grid cell as the coefficient of variation between the focal cell and its eight neighbors. This approach assumes that biogeographic boundaries often occur in areas with sharp climate transitions. Grid cells with higher heterogeneity values indicate they are different from neighboring ones. (2) Tectonic movement was determined by reconstructing the geographic locations of present-day grid cell centroids in timesteps of 1 Ma back to their historical positions 65 Ma. For each grid cell, we calculated the historical distance between a grid cell and eight neighboring cells at each timestep, and then computed the standard deviation of these distances across the last 65 million years, representing the variability of geographical distances between grid cells across time. (3) To estimate orographic barriers, we obtained elevation data from WorldClim v.2.1 and calculated the mean absolute difference in elevation between a focal grid cell and 8 neighboring cells. (4) We also included past climate velocity during the Quaternary at 5-arcmin resolution by extracting values across grid cells. We standardized all predictor variables to have a mean of 0 and variance of 1. All predictor variables showed variance inflation factors less than 2 indicating minimal collinearity. We hope that this approach has minimized the potential of circularity in our analysis. This information is now provided in Lines 390–403 of the revised manuscript.

Minor comments:

1. L65-66, maybe a good place to mention this study is based on occurrence records, rather than regional data, like previous studies.

RESPONSE 2.7: We thank the reviewer for this suggestion. Accordingly, we have revised wording to indicate that our study is based on occurrence records as follows:

“Specifically, we used species distribution models to analyze 402 million occurrence records for vascular plants resulting in individual species-level native range maps for 189,269 species under present and future climatic projections throughout the twenty-first century.”

2. L70, why the SSP126 is a “strong” scenario?

RESPONSE 2.8: We have revised wording to “*best-case*” scenario to improve clarity.

3. L105, Eremean or Tropics-Eremean?

RESPONSE 2.9: Thank you. Wording has been revised to “Tropics-Eremean” as suggested.

4. L113, could not see these results from Fig. S4.

RESPONSE 2.10: Following substantial revision of the manuscript to improve clarity and remove redundancy we have removed this analysis from the current manuscript without impacting the key findings that climate change can alter biogeographic regions.

5. L161, it seems not only “future” boundaries were analyzed here.

RESPONSE 2.11: We thank the reviewer for spotting this. We have accordingly revised heading to “**Determinants of biogeographical boundaries**” to indicate that both present-day and future boundaries were considered in the analysis.

6. In the materials and methods, P3, step v, “with known occurrences”. The known occurrences referred to the records obtained from the GBIF. P4 L2, what were the “17 communities matrices”? P15, it is a bit hard to understand what the response variable was in the biogeographic boundary regression. From the description, it seems the authors constructed a Y/N variable with cells overlapping with the boundary as 1, and all other grid cells (places in white in Fig. 3) as 0? I am wondering whether the huge amount of 0s would influence the analyses.

RESPONSE 2.12: The reviewer is correct that the known occurrences refer to the records obtained from GBIF. We then revised wording to improve clarity on our definition of community matrices for the different climate scenarios as follows:

“For each climate scenario, we overlaid each species modelled distributions onto equal area grid cells of 100 km × 100 km (Mollweide global projection system) to convert the predicted distribution data to a community matrix of 189,269 species × 14,810 grid cells using the polys2comm function in the R package phyloregion v.1.0.9 (ref.⁷⁸). This resulted in a total of 17 different community matrices of 189,269 species × 14,810 grid cells (one for the present condition and four for the four different future time horizons × four SSPs).”

Regarding our analysis of determinants of biogeographic boundaries, we have provided more clarity that we built our models in which the response variable was a binomially distributed binary variable determining whether a given cell is in contact with a biogeographic boundary or

not. Along these lines, for each time horizon and climate scenario, we split the vector polygons based on floristic regions, rasterized the positions of boundaries between floristic regions that were not separated by the sea, and generated buffers of 200 km around the boundary cells. The cells directly touching boundary lines were coded as “YES” and the remaining buffered cells not touching the boundary lines were coded “NO”. This information is now provided in the revised manuscript in Lines 419–428 as follows:

“Next, we used a simultaneous autoregressive spatial (SAR) model with binomial error distribution to assess relationships of the position of plant biogeographic boundaries and the predictors while accounting for spatial autocorrelation. This was achieved by building our models with the response variable being a binomially distributed binary variable determining whether a given cell is in contact with a biogeographic boundary or not. Along these lines, for each time horizon and climate scenario, we split the vector polygons based on floristic regions, rasterized the positions of boundaries between floristic regions that were not separated by the sea, and generated buffers of 200 km around the boundary cells. The cells directly touching boundary lines were coded as “YES” and the remaining buffered cells not touching the boundary lines were coded “NO”.”

7. In Fig.1, perhaps adding a symbol of a phylogenetic tree to "input data" will make it more intuitive.

RESPONSE 2.13: We thank the reviewer for this suggestion. We have accordingly added a symbol of a phylogenetic tree to the input data section which we believe improved the presentation of the figure. The revised figure is illustrated below.

Fig. 1. Hypotheses for how climate-induced biodiversity change can alter the future of floristic regions. Clusters of present floristic regions in non-metric multidimensional scaling (NMDS) space defined by clustering species occurrences with phylogenetic information and applying pairwise Simpson’s β -diversity, β_{sim} , where a is the number of shared species between two grid cells, and b and c are the numbers of species unique to each grid cell. As climate changes due to human impacts, species may undergo range shifts along environmental gradients in response to environmental factors such as temperature to maintain equilibrium with suitable living conditions. Consequently, and depending on the climate scenarios over the 21st century, we predict that climate change can homogenize, redefine or differentiate the future of floristic regions.

8. Fig. 4, it seems unclear why the elevation was significant in T2, rather than in other times.

RESPONSE 2.14: Following substantial revisions and improvements of the paper to improve clarity and remove redundancy, we have accordingly revised the analysis of determinants of biogeographical boundaries. We found that boundary shifts are projected to be driven by variations in past climate change during the Quaternary, altitude variation and transitions in temperature seasonality. This information is now provided as new Fig. 5 below.

Fig. 5. Determinants of plant biogeographic boundaries predicted by a hierarchical generalized model. The model incorporates predictors hypothesized to determine the positions of biogeographic boundaries including climatic heterogeneity (annual temperature, temperature seasonality, annual precipitation, and precipitation seasonality), orogeny (altitude variation), tectonic movements, velocity of past climate, and spatial autocovariates. Indicated are determinants of all biogeographic boundaries in present-day versus future climate scenarios (2021-2040, 2041-2060, 2061-2080, and 2081-2100) for all region boundaries (top row) and realm boundaries (bottom row). Standardized effect sizes are calculated using Fisher's z scores with 200 replicates, providing a quantitative representation of the impact of each predictor. The error bars are 95% confidence intervals of z. Statistically significant predictors are marked with asterisks (*), indicating a significance level of $P < 0.05$.

9. In supplementary lines 86-87, which tool was used for the tree construction?

RESPONSE 2.15: We have now provided more clarity on how the phylogenetic tree was generated as follows:

“We then matched the community data to the most updated phylogenetic tree of the world's vascular plants⁷⁹. The phylogenetic tree was generated using the R package V.PhyloMaker2 v.0.1.0 (73) with function phylo.maker based on the expanded megaphylogeny (GBOTB.extended.TPL) of ref.⁷⁹ as a backbone and the function build.nodes.1 in the R package V.PhyloMaker2 v.0.1.0 (73). Missing taxa from the megaphylogeny were added using V.PhyloMaker2 under scenario “S2” that allows generation of random trees. We generated one tree and used it to analyze compositional turnover (phylogenetic beta diversity) based on Simpson beta diversity index (β_{sim})⁸⁰.”

References:

Daru, B.H., Davies, T.J., Willis, C.G., Meineke, E.K., Ronk, A., Zobel, M., et al. (2021). Widespread homogenization of plant communities in the Anthropocene. *Nat. Commun.* 12, 1–10.

RESPONSE 2.16: We thank the reviewer for pointing us to this reference. In response to the reviewer's similar comment above, we used MaxEnt as the model of choice in the current analysis rather than an ensemble approach as in Daru et al. 2021 because our current study incorporated hundreds of thousands of species spanning five time horizons and four climate scenarios for a final dataset of 3,217,573 range maps. Thus, creating ensembles for each would be computationally expensive and time-consuming. Therefore, we decided to run each MaxEnt model five times and take the median as a more efficient approach.

Reviewer #3 (Remarks to the Author):

RESPONSE 3.1: We thank the reviewer for co-reviewing the manuscript and providing comments which we believed significantly improved the communication of the key findings.

REVIEWERS' COMMENTS

Reviewer #2 (Remarks to the Author):

I am delighted with the revisions of Manuscript #NCOMMS-24-04378A, Climate change alters the future of natural floristic regions of deep evolutionary origins. The authors have admirably addressed my concerns, impressively reanalysing such an extensive dataset within such a short period. The MS is now both highly readable and informative.

RESPONSE 1.1: We thank the reviewer for the positive remarks and for finding our revised manuscript readable and informative.

Below are some minor comments:

L138-139, not sure I understand this sentence, particularly about the sea.

RESPONSE 1.2: We have now revised this sentence to improve clarity by removing reference to the sea throughout the manuscript where necessary without impacting meaning in the original sentence as follows:

“We evaluated shifts in region and realm boundaries as the boundaries between biogeographic regions, at a grain resolution of 100 km.”

L148 & L 159, Fig. 3 should be Fig. 4.

RESPONSE 1.3: We thank the reviewer for drawing our attention to this. We have accordingly revised references to these sentences to correctly cite Fig. 4.

L178-182, I did not see a significant correlation between tectonic separation and present-day realm boundaries, in contrast, tectonics seems positively (although weakly) linked to realm boundaries in the time horizon of 2081-2100.

RESPONSE 1.4: We appreciate this comment and have thoroughly revised our interpretation of the analysis of determinants of realm boundaries as follows:

“...while temperature seasonality, precipitation seasonality, altitude variation, tectonic separation, and past climate change during the Quaternary are important in determining the positions of present-day realm boundaries, only temperature seasonality, altitude variation, and tectonic separation, will remain consistent in shaping future realm boundaries (Fig. 5).”

L187 and some other relevant places, what did the “abrupt transitions in climate seasonality” refer to?

RESPONSE 1.5: For simplicity, we have revised our usage of this term throughout the manuscript to refer to either variations in temperature seasonality or precipitation seasonality where relevant, and defined as the variability or fluctuation of temperature or precipitation over the course of a year.

L187-191, did the authors mean that the temperature seasonality only occurs in mountainous areas?

RESPONSE 1.6: We acknowledge that temperature seasonality can occur in various environments, including lowlands. Our intention was to highlight mountainous regions due to their unique topographic features such as elevation, slope, and aspect, which can in turn create pronounced variations and temperature and precipitation within short distances. This can lead to distinct phenological patterns and ecological niches, making mountainous areas particularly

sensitive to climate change. We have accordingly clarified this point in the revised manuscript to improve clarity as follows:

“Likewise, variations in temperature seasonality, particularly in mountainous regions, could disrupt phenological cycles (e.g., flowering, or leafing times), pushing species beyond their physiological tolerances. Thus, the physical barrier of mountains and the variation in temperature with altitude will likely continue to be a major factor shaping plant distributions in the coming decades.”

L191-193, many other studies (e.g., Leprieur et al., 2011; Svenning et al., 2011; Xu et al., 2023) have also revealed the importance of paleoclimatic legacy in shaping the spatial turnover of biodiversity, thus, it is recommended to add some references to strengthen this point.

References:

F. Leprieur, P. A. Tedesco, B. Hugueny, O. Beauchard, H. H. Durr, S. Brosse, T. Oberdorff, Partitioning global patterns of freshwater fish beta diversity reveals contrasting signatures of past climate changes. *Ecol. Lett.* 14, 325–334 (2011).

J. C. Svenning, C. Flojgaard, A. Baselga, Climate, history and neutrality as drivers of mammal beta diversity in Europe: Insights from multiscale deconstruction. *J. Anim. Ecol.* 80, 393–402 (2011).

Xu W-B, Guo W-Y, Serra-Diaz J, Schrod F, Eiserhardt W, Enquist B, Maitner B, Merow C, Violle C, Anand M et al. 2023. Global beta-diversity of angiosperm trees is shaped by quaternary climate change. *Sci. Adv.* 9: eadd8553.

RESPONSE 1.7: We thank the reviewer for pointing to the additional literature which we have enjoyed reviewing and have accordingly cited them in our paper as follows:

“These findings suggest that past climate changes have left lasting legacies that can shape the future redistribution of plant biogeographic region boundaries under anthropogenic change^{40–42}.”

L244, should be Table S4, and please check the citation numbers in the table.

RESPONSE 1.8: We appreciate the reviewer drawing our attention to this. We have accordingly revised our citation of this table to Supplementary Table 4. Our revision also ensures the citation numbers of the tables throughout the manuscript is consistent.

Ref. 88 and 89, are they the same?

RESPONSE 1.8: We have revised wording to remove the duplicated references.

Reviewer #3 (Remarks to the Author):

RESPONSE 2.1: We thank the reviewer for co-reviewing the manuscript and providing the listed reports which have helped improve the communication of the key findings of this manuscript that climate change can influence plant biogeographic regions.